# Contemporary Management of Postoperative Crohn’s Disease after Ileocolonic Resection

**DOI:** 10.3390/jcm11226746

**Published:** 2022-11-15

**Authors:** Jurij Hanzel, David Drobne

**Affiliations:** 1Department of Gastroenterology, University Medical Center Ljubljana, 1000 Ljubljana, Slovenia; 2Department of Internal Medicine, Faculty of Medicine, University of Ljubljana, 1000 Ljubljana, Slovenia

**Keywords:** surgery, prophylaxis, fecal calprotectin, magnetic resonance enterography, intestinal ultrasound, endoscopy, noninvasive monitoring

## Abstract

Surgery remains an important treatment modality in the multidisciplinary management of patients with Crohn’s disease (CD). To illustrate the recent advances in the management of postoperative CD we outline the contemporary approach to treatment: diagnosing disease recurrence using endoscopy or noninvasive methods and risk stratification underlying decisions to institute treatment. Endoscopic scoring indices are being refined to guide treatment decisions by accurately estimating the risk of recurrence based on endoscopic appearance. The original Rutgeerts score has been modified to separate anastomotic lesions from lesions in the neoterminal ileum. Two further indices, the REMIND score and the POCER index, were recently developed with the same intention. Noninvasive monitoring for recurrence using a method with high negative predictive value has the potential to simplify management algorithms and only perform ileocolonoscopy in a subset of patients. Fecal calprotectin, intestinal ultrasound, and magnetic resonance enterography are all being evaluated for this purpose. The use of infliximab for the prevention of postoperative recurrence is well supported by data, but management decisions are fraught with uncertainty for patients with previous exposure to biologics. Data on the use of ustekinumab and vedolizumab for postoperative CD are emerging, but controlled studies are lacking.

## 1. Introduction

Crohn’s disease (CD) is a chronic progressive inflammatory bowel disease leading to bowel damage and disability [1]. The expanding treatment armamentarium and wider use of biologics has paralleled a decline in the rates of surgical resection [2,3]. Nonetheless, surgery remains an important treatment modality in patients with obstructive symptoms, penetrating complications, or medically refractory disease. Up to 25% of patients with CD undergo surgery within five years of diagnosis [4], and up to a quarter of these patients will undergo a second resection within five years of the first [5]. Managing post-operative CD is challenging and complex as many of its aspects are incompletely supported by evidence.

Society guidelines suggest stratification on clinical risk factors to guide prophylactic treatment after resection, followed by endoscopy at 6 months to inform potential treatment escalation [6,7,8]. Notably, risk factors for recurrence have never been prospectively validated and guidelines differ in the definition of a patient at high risk for recurrence, with the British Society of Gastroenterology requiring the presence of at least two risk factors [8], whilst the American Gastroenterological Association and the European Crohn’s and Colitis Organization mandate the presence of a single risk factor [6,7]. Endoscopy-based management is supported by the findings of the Post-Operative Crohn’s Endoscopic Recurrence (POCER) study, a randomized trial demonstrating the superiority of early colonoscopy compared to standard care [9]. In spite of the crucial role of endoscopy in the management of postoperative CD, no endoscopic index has been fully validated for this purpose, including the most widely used Rutgeerts score [10]. Further uncertainty surrounds the choice of therapeutic agent to prevent postoperative recurrence. Metronidazole and thiopurines continue to be used despite marginal efficacy over placebo and the potential for serious adverse events [11]. Although treatment with infliximab led to a statistically significant 28.9% reduction in the risk of endoscopic recurrence, the PREVENT trial did not meet its primary endpoint composed of clinical and endoscopic recurrence, as only 18.1% (20/110) of patients with endoscopic recurrence as defined by the Rutgeerts Score ≥ i2 also had recurrence based on the CD Activity Index (CDAI) (defined by a total CDAI score >200 and a ≥70-point increase from baseline) [12]. Consequently, none of the drugs used for the treatment of CD have received regulatory approval for prevention of postoperative recurrence.

The areas of uncertainty outlined above have fueled intense research efforts in recent years. These focused on refining existing endoscopic indices, the development of novel endoscopic indices, the use of biomarkers to support noninvasive diagnosis of postoperative recurrence, the utility of cross-sectional imaging to replace endoscopy, and the use of biologics beyond tumor necrosis factor (TNF)-α inhibitors to prevent postoperative recurrence. We searched Pubmed from inception to November 2022, with a focus on studies published after 2015 with the following search strategy: ((post-operative [Title/Abstract]) OR (postoperative [Title/Abstract]) OR (post-surgical [Title/Abstract]) OR (postsurgical [Title/Abstract]) OR (resection [Title/Abstract]) OR (recurrence [Title/Abstract])) AND ((Crohn [Title/Abstract]) OR (Crohn’s [Title/Abstract])). We identified publications pertaining to (1) endoscopic assessment of postoperative CD; (2) noninvasive methods of diagnosing postoperative recurrence (biomarkers, imaging); (3) medical prophylactic treatment. Reference lists of included publications were searched to identify potential additional relevant studies.

In this narrative review we aim to provide an overview of recent developments in the diagnosis of postoperative CD through endoscopy, biomarkers, and imaging, as well as advances in its medical treatment.

## 2. Diagnosing Recurrence in Postoperative Crohn’s Disease

### 2.1. Endoscopy

Endoscopic assessment is the gold standard for diagnosing postoperative recurrence and is the cornerstone of decision-making in the postoperative period. A colonoscopy-based monitoring strategy was evaluated in the randomized POCER trial and was shown to be superior to conventional management in reducing the rate of recurrence at 18 months [9]. There is some disagreement as to what constitutes endoscopic recurrence, namely whether lesions confined to the anastomosis carry the same prognostic significance as lesions in the neoterminal ileum. In addition to the Rutgeerts score, two new endoscopic indices have recently been developed. A comparison of available endoscopic indices is presented in Figure 1.

#### 2.1.1. (Modified) Rutgeerts Score

Society guidelines advocate endoscopic evaluation in all patients at 6 months after surgery. Endoscopic disease activity in the neoterminal ileum and the ileocolonic anastomosis has traditionally been evaluated using the Rutgeerts score (Table 1), where i1–i4 is considered endoscopic recurrence and escalation of therapy is recommend for scores of i2 an higher [10]. Despite its widespread use in clinical practice and clinical trials, the score’s operating characteristics have not been fully studied [13], with its responsiveness remaining unknown. The inter-rater reliability was shown to be “substantial” upon evaluation by expert endoscopists, although defining aphthous ulcers in the neoterminal ileum was a source of disagreement, potentially due to difficulty of separating small ulcers from mucus or residual debris.

The hypothesis than anastomotic lesions portend a better prognosis compared to lesions in the neoterminal ileum resulted in the modified Rutgeerts score [14], which separates i2 into isolated lesions confined to the ileocolonic anastomosis (i2a), while all other lesions qualifying for i2 on the original score are classified as i2b (>5 aphthous ulcers or large lesions, with normal mucosa in-between, in the neo-terminal ileum, regardless of concomitant anastomotic lesions) (Table 1). Nonetheless, recent research has demonstrated histological features of CD, rather than ischemia, in the majority of anastomotic ulcers [17].

Comparisons between the original Rutgeerts score and its modification have yielded conflicting results for clinical outcomes [15,18,19,20,21]. The discrepancies could perhaps be explained by retrospective design of most studies and endoscopic assessment based on still images. The only prospective study evaluating the association of the modified Rutgeerts score with subsequent clinical outcomes was a French cohort study of 225 patients (193 with long-term follow-up) with local endoscopic reading [15]. The study indicated an incremental prognostic benefit of the modified score as clinical recurrence-free survival was similar between i0 and i2a, but significantly shorter for i2b compared to i0. These findings thus support the reasoning that isolated anastomotic ulcers have a better prognosis than ulcers in the neoterminal ileum.

In a bicentric retrospective study, patients with i2a and i2b did not differ in the rate of clinical or surgical postoperative recurrence [19]. In contrast to these findings, a single-center retrospective study from Chicago failed to find a statistically significant difference for endoscopic disease progression (defined as progression to i3 or i4) in patients with i2a lesions compared to patients with i0/1 lesions [18]. The comparison may have been underpowered as rates of endoscopic progression were numerically higher for i2a (hazard ratio [HR] 2.30; 95% confidence interval [CI] 0.80–6.66). Similar trends were observed for surgical recurrence: rates were unequivocally higher for i2b in comparison to i0/1 (HR 3.64; 95% CI 1.10–12.1), but only numerically higher for i2a (HR 1.43; 95% CI 0.35–5.77) with broad confidence intervals.

A retrospective study from Cleveland lends further credibility to the hypothesis that i2a may in fact confer a higher risk of recurrence compared to i0/1, but numerically smaller and thus more likely to result in a non-significant result than i2b [21]. In this study, the adjusted odds ratio (OR) of endoscopic progression to i3/4 was 5.53 (95% CI 2.50–12.77) for i2b compared to i0/1 and 2.11 (95% CI 0.89–4.97) for i2a. A sensitivity analysis defining endoscopic progression as ≥i2b indicated that the risk was indeed higher for i2a lesions in comparison to i0/1. The study had a low prevalence of surgical recurrence, 13.6% (27/199), and was underpowered to detect an association between the endoscopic score and risk for second surgery. A further retrospective study found that anastomotic ulcers were common, occurring in 52.2% of patients after surgery, and associated with an increased risk of a composite endpoint of surgical and endoscopic (i2b or higher) recurrence [20].

Recently, an individual patient data meta-analysis of the abovementioned studies and some smaller series was published [22]. It included 400 patients with i2 scores and compared the risk of clinical and surgical postoperative recurrence between patients with i2a and i2b scores. There was no significant difference between the two groups, although the risk for clinical postoperative recurrence was numerically higher in patients with i2b lesions and the difference increased with time (9% vs. 11% at 1 year, 33% vs. 25% at 3 years, and 47% vs. 36% at 5 years). Rates of surgical recurrence were low in all studies (up to 6% at 5 years) and not even numerical differences were observed between groups. No comparisons between i1 and i2 scores were made. The authors concluded that the same treatment strategy could be used for both groups given the comparable recurrence rates even though the optimal strategy remains to be determined.

Taken together, the prognostic impact of anastomotic ulcers remains ambiguous and it is unclear how they should be incorporated in clinical decision-making. It seems plausible that anastomotic ulcers confer a higher risk of recurrence than observed with i0/1, but numerically smaller than with i2b, with the difference appearing nonsignificant in studies with fewer than 250 participants. It should also be noted that studies with endoscopic endpoints [18,21] are both more sensitive (a substantial proportion of patients with endoscopic recurrence are asymptomatic) and specific (not all diarrhea in the postoperative setting reflects active disease) than studies with clinical endpoints [15,19] and thus probably more accurately reflect the significance of separating i2 lesions. Finally, it should be borne in mind that the POCER study, the only strategic randomized trial supporting current management of postoperative CD, used the original Rutgeerts score where anastomotic ulcers already constituted recurrence and led to treatment escalation in the active care arm [9]. A prospective randomized trial of patients with i2 endoscopic recurrence at 6 months is ongoing in France (POMEROL; NCT05072782). Patients will be randomized to either receive infliximab (intravenous induction, followed by subcutaneous maintenance) or continue with an unchanged treatment regimen (either no treatment or continued prophylaxis with thiopurines or methotrexate) and assessed endoscopically at 12 months for the primary endpoint of a Rutgeerts score of i0/1.

#### 2.1.2. REMIND Score

The REMIND score was developed in a French multicentric prospective study mentioned above [15]. This score separates anastomotic lesions (sub-score A) from ileal lesions (sub-score I), with anastomotic lesions graded based on their circumferential extent and ileal lesions as defined by the original Rutgeerts score (Table 1). The main finding of the study was that long-term outcomes were dependent on ileal, rather than anastomotic, lesions. Only the most severe anastomotic lesion, anastomotic stenosis, was associated with subsequent occlusive complications, but not clinical recurrence. A notable finding of the study was the high clinical recurrence rate in patients with I(1) lesions that did not differ significantly from recurrence rates with more severe ileal lesions. In summary, results from the REMIND cohort suggest that a lower threshold for escalating treatment should be applied to ileal lesions with treatment escalation at i1, rather than i2, while the presence of anastomotic lesions is a minor factor in the decision process.

Although good inter-rater reliability (weighted kappa coefficient of 0.82) was demonstrated in the original study, the score requires further validation in independent cohorts [23], particularly regarding its impact on treatment decisions in comparison with the Rutgeerts score.

#### 2.1.3. POCER Index

The POCER index was developed on the subset of patients from the active arm of the POCER trial who had endoscopic assessment at 6 and 18 months (*n* = 85) [16]. Five new scoring items evaluated at the anastomosis were selected a priori to be assessed for their association with subsequent endoscopic recurrence (defined as a Rutgeerts score of i2 or greater): (1) total number of ulcers at the anastomosis; (2) ulcer depth; (3) circumferential extent of ulcers; (4) size of the largest ulcer; (5) presence of stenosis. None of the items were associated with subsequent endoscopic recurrence in isolation, but the anastomotic ulcer depth and circumference were selected to develop the new index based on factor analysis (Table 1).

Interestingly, neoterminal ileal lesions were not associated with subsequent recurrence in this population. The association of the POCER index with clinical recurrence or need for surgical reintervention remains to be studied and the index requires validation in an independent cohort. Although ulcer depth was measured using a standard biopsy forceps, assessing depth can be challenging in practice, as has been shown with the CD endoscopic index of severity (CDEIS) [24].

### 2.2. Fecal and Serum Biomarkers

Despite being the gold standard, colonoscopy is invasive, requires bowel preparation and is not without risk. In fact, patients rated colonoscopy as the least acceptable monitoring tool [25]. In contrast, stool sampling, and, to an even greater extent, serum sampling are well accepted by patients and hold promise to be able to stratify patients by the risk for recurrence and individualize referrals for colonoscopy.

#### 2.2.1. Fecal Biomarkers

Fecal calprotectin is a calcium- and zinc-binding protein expressed by neutrophils that is widely used for the noninvasive monitoring of CD [26]. Given that histologic changes preceding subsequent endoscopic recurrence are known to develop within days of surgery [27], fecal calprotectin could not only serve as a diagnostic biomarker (Does this patient have endoscopic recurrence?), but also a predictive biomarker (Will this patient develop endoscopic recurrence?).

Its performance in postoperative CD was evaluated by two meta-analyses [28,29]. Both meta-analyses defined a Rutgeerts score of ≥i2 as endoscopic recurrence. At a cutoff of 100 mcg/g, the sensitivity for endoscopic recurrence was 81% and the specificity 57%, at 150 mcg/g, the sensitivity was 70% and specificity 69% [29]. A positive association between calprotectin concentrations and the severity of endoscopic recurrence has been demonstrated [30]. By extension, in a study evaluating the performance of fecal calprotectin against both versions of the Rutgeerts score showed superior test characteristics (cutoff 100 mcg/g; sensitivity: 74% vs. 48%; specificity: 91% vs. 33%) with the modified Rutgeerts, reflecting the fact that calprotectin concentrations were lower in patients with i2a than i2b [31].

The utility of fecal calprotectin is limited by the variation between assays, diurnal variation and the moderate sensitivity and specificity. Some of these limitations may be overcome by serial measurements. This practice is supported by two small studies, which indicate that this approach merits further research. In a prospective French study of 48 patients, the kinetics of fecal calprotectin, but not absolute values, in the first three months of surgery were associated with subsequent endoscopic recurrence [32]. An increase of fecal calprotectin >10% from baseline (i.e., within 21 days after surgery) to month 3 had a 78.6% positive predictive value for endoscopic recurrence at 6 months. These early changes could be used to identify patients at high risk for endoscopic recurrence and expedite endoscopic assessment. Yamamoto et al. [33] studied patients who had no endoscopic recurrence at 6 months and underwent fecal calprotectin sampling every 2 months for up to 24 months. Of the patients with calprotectin below 140 mcg/g throughout follow-up, only 9% (2/22) developed endoscopic recurrence, whilst 75% (6/8) of patients with at least one measurement above 140 mcg/g had endoscopic recurrence. This study highlights the potential for a noninvasive follow-up approach after the index colonoscopy—a period for which the optimal management strategy is unknown.

#### 2.2.2. Serum Biomarkers

The Endoscopic healing index (EHI) is a recently validated assay measuring 13 serum proteins to noninvasively identify patients with CD in endoscopic remission (Simple endoscopic score for CD [SES-CD] ≤ 2) [34]. At a cutoff value of 20 points (calculated by a proprietary algorithm), the performance of EHI was similar to that of fecal calprotectin in the training and validation cohorts.

The EHI was measured for stored serum samples from the POCER trial [35]. At 6 months, an EHI ≤ 20 had a negative predictive value of 75.7% for endoscopic recurrence. At this time point, both fecal calprotectin and EHI performed similarly [35,36]. At 18 months postoperatively, however, the EHI could not discriminate between remission and recurrence, unlike fecal calprotectin which maintained a negative predictive value of 89.7% for a cutoff of 100 mcg/g. The cause for this discrepancy at 18 months is unknown and may potentially be related to the fact that the EHI was developed using the SES-CD and not the Rutgeerts score: a single aphthous ulcer in the neoterminal ileum (i1) would score 3 points on the SES-CD, as would 6 aphthous ulcers (i2), provided that the percentage of ulcerated or affected surface was below 10% and 50%, respectively. The authors also explored the possibility of using both tests in tandem, which resulted in a modest improvement in test characteristics. It is thus unclear whether performing both tests simultaneously improves diagnostic performance to a meaningful extent.

A group of Spanish investigators found that measuring serum cytokines, namely interleukin-6 and interferon-γ, together with fecal calprotectin increased the diagnostic accuracy for endoscopic recurrence [37]. Admittedly, the incremental value of measuring serum cytokines was modest as the area under the receiver operating characteristic curve increased from 0.88 to 0.90 after their inclusion in the model.

### 2.3. Cross-Sectional Imaging

With CD being a transmural disease, there is a lingering concern that endoscopic evaluation limited to the mucosa is inadequate to account for the full spectrum of disease, overlooking changes in the intestinal wall that could affect subsequent management. Cross-sectional imaging has the potential to overcome this limitation; moreover, it is noninvasive and neither ultrasound nor magnetic resonance imaging expose patients to ionizing radiation, making it an attractive monitoring tool for postoperative CD.

#### 2.3.1. Intestinal Ultrasound

DiCandio et al. [38] were the first to use intestinal ultrasound to diagnose postoperative recurrence of CD in 1986—four years before the publication of the Rutgeerts score. Different ultrasonographic techniques have been used: bowel sonography without the use of intravenous or oral contrast, small intestine contrast ultrasound with the use of oral contrast solution, and contrast-enhanced ultrasound using an intravenous contrast medium. Oral contrast solution serves to facilitate assessment by distending bowel loops, while intravenous contrast enables the assessment of vascularization and hyperemia in active CD.

A systematic review of ten studies including 536 patients found that the overall sensitivity of intestinal ultrasound, pooling all three techniques, was 94% and specificity 84% [39]. Small intestine contrast ultrasound had a higher sensitivity (99% vs. 82%), but lower specificity (74% vs. 88%) compared to simple bowel sonography. The cutoff value of bowel wall thickness to diagnose recurrence was almost universally set to 3 mm—following established conventions for intestinal ultrasound [40]. In the systematic review, a single study evaluated contrast-enhanced ultrasound, demonstrating a sensitivity of 90% and specificity of 82% [41]. Since the publication of the systematic review, one further study on the use of this technique was conducted, where the accuracy of bowel sonography without contrast and contrast-enhanced ultrasonography for diagnosing endoscopic recurrence was identical—90.7% [42]. The added value of intravenous contrast lay in the identification of severe recurrence (≥i3). The high prevalence of endoscopic recurrence in the cohort, 83.3% (90/108), should be borne in mind when interpreting the results of this study.

#### 2.3.2. Magnetic Resonance and Computed Tomography Enterography

The sensitivity of enterography to detect endoscopic recurrence has ranged from 92 to 96% and its specificity from 75 to 88% [43,44,45,46]. In line with the notion that mucosal visualization during colonoscopy provides an incomplete appraisal of disease burden, a recent study explored the concordance between radiographic and endoscopic findings [47]. In this retrospective cohort study, the images of 216 postoperative patients with enterography and colonoscopy performed within 90 days of each other were reviewed. Endoscopic recurrence was defined as ≥i2b. The majority of patients, 54.2% (117/216), had concordant findings between radiology and endoscopy, 41.7% (90/216) had radiological, but not endoscopic signs of active disease, and 4.2% (9/216) had endoscopic, but not radiological signs of active disease. Notably, patients with radiological, but not endoscopic, disease activity had a shorter time to endoscopic recurrence and greater risk of surgical recurrence.

These findings seem concerning, as they suggest that endoscopic assessment systematically underestimates the risk for recurrence, thereby questioning the validity of a monitoring approach based on endoscopy. The majority of discrepant results are readily explained by the cut-off for endoscopic recurrence of i2b in the study: of the 90 patients with radiologic, but not endoscopic, signs of recurrence, 62.2% (56/90) had endoscopically active disease which did not fulfil criteria for recurrence in the study (43 patients with i2a; 13 patients with i1). Proximal small bowel disease was the reason for discordant findings in only three patients. In a sensitivity analysis, where the threshold for endoscopic recurrence was set at i2a, there was no longer a significant difference of subsequent endoscopic recurrence between patients with no radiologic or endoscopic signs of recurrence (46.2%) and patients with radiologic, but not endoscopic, signs of recurrence (55.6%). In summary, the results of this study highlight the gradient of risk for recurrence from i1, across i2a to i2b, rather than an important intrinsic difference between radiologic and endoscopic monitoring strategies that would lead to consequences for patient management.

Recently, the Magnetic Resonance Imaging in CD to Predict Postoperative Recurrence (MONITOR) index for the systematic evaluation of magnetic resonance enterography (MRE) in the setting of postoperative CD was developed and partly validated [48]. The index was developed on 73 paired endoscopic and MRE assessments in a French tertiary center. Seven items with good intra- and inter-rater reliability were included: wall thickening, contrast enhancement, T2 signal increase, diffusion-weighted signal increase, edema, ulcers, and the length of the diseases segment (<20 mm versus ≥20 mm). Ulcers are scored with 2.5 points, while the presence of one of the six other items scores one point each. The optimal cut-off for endoscopic recurrence defined as >i1 was 1 point, yielding an area under the receiver operating characteristic curve of 0.80 and a sensitivity of 79%, specificity of 55%, positive predictive value of 68%, and negative predictive value of 68%. The operating characteristics of the index in a validation cohort of 17 patients were largely similar.

This is the first MRE index designed specifically for postoperative CD and is a considerable step towards noninvasive monitoring of patients. The negative predictive value, however, is not high enough to confidently identify patients at low risk of recurrence who can forego endoscopy. The index is pending further validation and assessment whether its utility can be increased by combining it with measurements of fecal calprotectin.

### 2.4. Novel and Emerging Biomarkers

Emerging biomarkers for the prediction and diagnosis of postoperative recurrence include single nucleotide polymorphisms (SNPs), transcriptomics, metabolomics and microbial markers. These could facilitate the decision to institute postoperative prophylactic therapy.

In a retrospective cohort of 372 patients with CD undergoing surgery, a polymorphism in transcription factor 4 (TCF4) conferred a significant risk of surgical recurrence (OR 4.10; 95% CI 2.37–7.11) [49]. In a study of 60 patients, RNA was extracted from the noninflamed ileal margin of resection specimens, the transcripts were later classified by random forest, a machine learning algorithm, to identify patients with i0 endoscopic scores [50]. In anti-TNF naïve patients, a clear transcriptional cluster separating patients with i0 scores from other patients was identified. In anti-TNF exposed patients, no association between transcriptional profiles and endoscopic scores were found. The investigators developed an ad hoc score to define an indolent disease course after surgery, which was associated with distinct transcriptional profiles even in anti-TNF experienced patients. In a small prospective study of 38 patients, elevated urinary levoglucosan concentrations were associated with endoscopic recurrence [51]. Levoglucosan concentrations were a diagnostic (i.e., recurrence had already occurred), rather than predictive, biomarker and it remains to be determined whether this biomarker offers an advantage compared to fecal calprotectin. In a prospective study of 121 patients undergoing ileocecal resection, fecal samples were collected at 1, 3, and 6 months postoperatively to characterize the microbiota [52]. In addition to this, the mucosa-associated microbiota was studied on biopsy samples. Both the mucosa-associated and fecal microbial profiles were superior to clinical factors in predicting endoscopic recurrence. The most significant change in patients with postoperative recurrence was the increased abundance of Fusobacteria.

All the emerging biomarkers outlined above require further validation in larger independent cohorts. Risk stratification immediately after resection is probably the largest unmet need and could potentially be refined with the use of these novel biomarkers.

## 3. Treatment of Postoperative Crohn’s Disease

The decision to start medical treatment after surgery is based on clinical risk factors for recurrence or on endoscopic evaluation at 6 months if postoperative prophylaxis was not immediately indicated. Risk factors for recurrence include: active smoking, prior intestinal resection, granulomas or myenteric plexitis in the resection specimen, penetrating disease, the presence of perianal disease, extensive (≥50 cm) small bowel disease, and age ≤30 years [6,7,8]. Risk factors mostly overlap between the different guidelines, but there is uncertainty about the risk threshold for starting medical prophylaxis. The disagreement underscores the fact that risk factors have not been validated prospectively and that there is uncertainty as to their relative contributions to recurrence risk. This is reflected by results of recently performed cohort studies where the association between the number of risk factors, risk of recurrence and benefit of prophylactic therapy was unpredictable at best [53,54,55].

No drug has regulatory approval specifically for the prevention of postoperative CD recurrence. This entity is not even explicitly mentioned in regulatory guidelines, but a coprimary end point of symptomatic and endoscopic remission is mandated for the registration of new medicinal products for the treatment of CD [56].

Notwithstanding the absence of regulatory approval, infliximab has the strongest evidence for preventing postoperative recurrence. It was evaluated in the PREVENT trial where patients with at least one risk factor for recurrence were randomized to receive infliximab (5 mg/kg every 8 weeks without the usual induction sequence) or placebo within 45 days of surgery. The primary endpoint was clinical recurrence, defined as a composite outcome consisting of a CD Activity Index (CDAI) score >200 and a ≥70-point increase from baseline, and endoscopic recurrence (Rutgeerts score ≥i2, determined by a central reader) or development of a new or re-draining fistula or abscess, before or at week 76. Endoscopic recurrence was a secondary outcome. Fewer patients in the infliximab group had clinical recurrence compared to the placebo group, but the difference was not statistically significant (12.9% [19/147] vs. 20.0% [30/150]; *p* = 0.097). The comparison for endoscopic recurrence showed superiority of infliximab (22.4% [33/147] vs. 51.3% [77/150]; *p* < 0.001). Only 18.1% (20/110) of patients with endoscopic recurrence also had recurrence based on the CDAI, which emphasizes the limitations of symptom-based scores in the postoperative setting. Additional elements of the trial design may have contributed to its result: infliximab was given without induction regimen (doses at weeks 0, 2, and 6), combination therapy with immunosuppressants was optional, and about 10% of patients had previously been exposed to infliximab. These factors may have increased the risk of immunogenicity and subsequent failure of infliximab: anti-drug antibodies were detected in 16.2% (all without immunosuppressants). Given that a drug-sensitive assay was used, the true prevalence of immunogenicity was likely underestimated.

5-aminosalicylates, nitroimidazole antibiotics, and thiopurines were also evaluated in the postoperative setting. According to a recent network meta-analysis, TNF antagonists and thiopurines (both alone and in combination with nitroimidazole antibiotics) were superior to placebo in preventing endoscopic recurrence, while nitroimidazole antibiotics in monotherapy and 5-aminosalicylates were no better than placebo. TNF antagonists were superior to thiopurines [11].

Given their lower efficacy and the absence of a clear advantage in safety, it is somewhat surprising that guidelines suggest the choice between thiopurines and TNF antagonists. In the randomized placebo-controlled TOPPIC trial of mercaptopurine, was superior to placebo in preventing clinical recurrence only in a subgroup analysis of smokers, but not the entire trial population [57]. These results were further supported by a recent individual patient data meta-analysis of six studies comparing TNF antagonists with thiopurines for postoperative CD [58]. Anti-TNF-α agents were superior to thiopurines for the prevention of endoscopic and clinical recurrence both in low- and high-risk patients.

A minority of patients in PREVENT, 22.6%, had been previously exposed to anti-TNF agents [12]. With the increasing number of treatment-refractory patients undergoing surgery after having failed multiple biologics, the choice of postoperative treatment will no longer be straightforward. A real-world study from Japan indicated that anti-TNF agents were less effective for the prevention of surgical recurrence in patients with previous exposure to biologics [59], although this observation is not universal [60]. Data on the effectiveness of non-anti-TNF biologics for the prevention of postoperative recurrence are are accumulating (Table 2) [61,62,63,64]. It should be noted that patients receiving ustekinumab or vedolizumab had almost universally been previously treated with biologics, most commonly at least one anti-TNF agent, and no degree of statistical adjustment can fully resolve the potential for residual confounding. In general, recurrence rates were numerically higher with vedolizumab and ustekinumab than with anti-TNF agents, although the difference was not always statistically significant. In a French study, ustekinumab was compared to azathioprine using propensity score matching and was found to be associated with lower rates of endoscopic recurrence [62]. A placebo-controlled randomized trial is ongoing for vedolizumab (REPREVIO; EudraCT 2015-000555-24). Previous exposure to anti-TNF agents is not an exclusion criterion, so the data are expected to be informative for daily clinical practice. Further prospective data on the use of vedolizumab and ustekinumab from larger cohorts are awaited to define their role in the postoperative setting.

## 4. Conclusions

Management of postoperative CD remains challenging: the need for surgery defines patients with higher risk for complications and a more aggressive disease which is less likely to respond to medical therapy. Unmet needs exist at all stages of management—risk stratification for institution of prophylaxis, diagnosing and defining postoperative recurrence, and the optimal use of biologics, particularly in patients with previous treatment failure (Table 3). Figure 2 outlines the current approach to postoperative CD with highlighted areas with the potential to change in the near future. Significantly more high-quality studies are needed to explore the efficacy of evolving approaches in the management of postoperative CD.

## Figures and Tables

**Figure 1 jcm-11-06746-f001:**
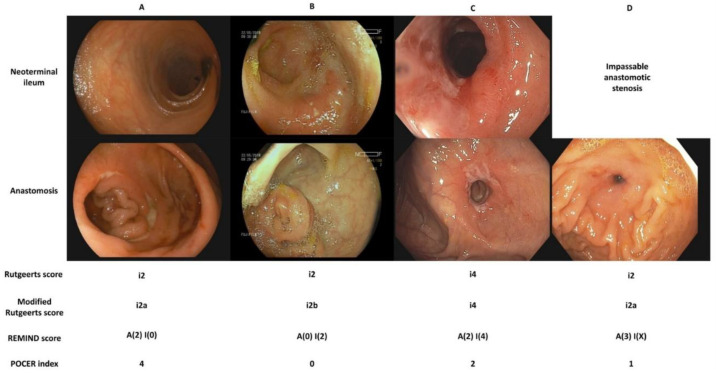
Comparison of endoscopic indices for the assessment of postoperative Crohn’s disease. (**A**). The neoterminal ileum is free of ulceration, two ulcers, one of them deeper than 2 mm, are present at the anastomosis and cover more than 50% of the circumference. (**B**). There are more than five aphthous ulcers with normal intervening mucosa in the neoterminal ileum. The anastomosis is free of ulceration. (**C**). The neoterminal ileum is diffusely inflamed with large ulcers. The anastomosis is superficially ulcerated along more than 50% of its circumference. (**D**). The anastomosis is impassable due to stenosis. A superficial ulcer covers less than 25% of its circumference. Note that an anastomotic stenosis should be scored as i2 on the Rutgeerts score and i2a on its modified version. Only a stenosis in the neoterminal ileum should be scored as i4.

**Figure 2 jcm-11-06746-f002:**
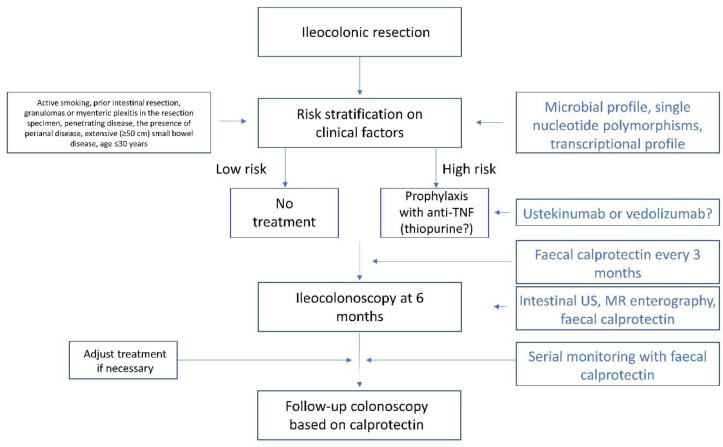
Proposed management algorithm for Crohn’s disease after ileocolonic resection, based on current guidelines (American Gastroenterological Association, British Society for Gastroenterology, European Crohn’s and Colitis Organization). Text in blue denotes potential changes to the algorithm in the near future. Abbreviations: MR—magnetic resonance; TNF—tumor necrosis factor; US—ultrasound.

**Table 1 jcm-11-06746-t001:** Endoscopic indices used for the assessment of postoperative Crohn’s disease.

**Rutgeerts Score** [10]	
i0	No lesions
i1	≤5 aphthous lesions in the neoterminal ileum
i2	>5 aphthous lesions with normal intervening mucosa or skip area of large lesions or lesions confined to the ileo-colonic anastomosis
i3	Diffuse aphthous ileitis with diffusely inflamed mucosa
i4	Large ulcers with diffuse mucosal inflammation or nodules or stenosis in the neo-terminal ileum
**Modified Rutgeerts Score** [14]	
i0	No lesions
i1	≤5 aphthous lesions in the neoterminal ileum
i2a	Lesions confined to the ileo-colonic anastomosis (including anastomotic stenosis)
i2b	>5 aphthous ulcers or large lesions, with normal mucosa in-between, in the neo-terminal ileum (with or without anastomotic lesions)
i3	Diffuse aphthous ileitis with diffusely inflamed mucosa
i4	Large ulcers with diffuse mucosal inflammation or nodules or stenosis in the neo-terminal ileum
**REMIND Score** [15]	
Anastomotic lesions (<1 cm in length after the anastomosis	
A (0)	No lesions
A (1)	Ulcerations covering less than 50% of the anastomosis circumference
A (2)	Ulcerations covering more than 50% of the anastomosis circumference
A (3)	Anastomotic stenosis
Ileal lesions	
I (0)	No lesions
I (1)	≤5 aphthous lesions in the neoterminal ileum
I (2)	>5 aphthous lesions with normal intervening mucosa or skip areas of larger lesions
I (3)	Diffuse aphthous ileitis with diffusely inflamed mucosa
I (4)	Diffuse inflammation with larger ulcers
**POCER Index** [16]	
0	No anastomotic ulcers
1	Superficial anastomotic ulcers (<2 mm in depth), <25% circumferential extent
2	Superficial anastomotic ulcers (<2 mm in depth), ≥25% circumferential extent
3	Deep anastomotic ulcer (≥1 ulcer with ≥2 mm depth), <25% circumferential extent
4	Deep anastomotic ulcer (≥1 ulcer with ≥2 mm depth), ≥25% circumferential extent

**Table 2 jcm-11-06746-t002:** Postoperative recurrence rates with ustekinumab and vedolizumab. Unless otherwise stated, recurrence rates were assessed between 6 and 12 months postoperatively.

	Endoscopic Recurrence (%)	
Study	Vedolizumab	Ustekinumab
Yamada et al. [61]	17/22 (75)	NA
Buisson et al. [62]	NA	9/32 (28)
Axelrad et al. [63] ^1^	13/27 (48)	10/28 (36)
Yanai et al. [64]	13/39 (33)	21/34 (62)

^1^ Median follow-up of 29 months. Figures combine endoscopic and radiographic recurrence.

**Table 3 jcm-11-06746-t003:** Current state of knowledge and future perspectives in the management of postoperative Crohn’s disease. Abbreviation: MRE—magnetic resonance enterography; TNF—tumor necrosis factor.

Aspect of Management	Current State of Knowledge	Future Perspectives
Risk stratification	Stratification on clinical and histological risk factors Not prospectively validated Uncertainty about the risk threshold for instituting prophylaxis	Prospective risk factor validation Development of refined clinical prediction models Risk-benefit analysis of upfront prophylaxis vs. step-up treatment in prospective cohorts
Diagnosing recurrence	Ileocolonoscopy at 6–12 months as the gold standard Endoscopic assessment using the Rutgeerts score Uncertainty about the prognostic implications of anastomotic lesions Uncertainty about the performance of emerging endoscopic indices (POCER, REMIND) Emerging role of noninvasive diagnostic modalities for diagnosing recurrence (stool- and serum-based biomarkers, MRE, intestinal ultrasound)	Defining the prognosis of lesions in the neoterminal ileum vs. lesions confined to the anastomosis in large prospective cohorts Comparison of the impact of different endoscopic indices on treatment outcomes Comparison of performance of endoscopic indices with different configurations of surgical anastomoses Full validation of POCER and REMIND indices Development of noninvasive monitoring algorithms with less reliance on endoscopy
Treatment	No drug has regulatory approval for the prevention of postoperative recurrence of Crohn’s disease No dedicated regulatory guidance for clinical trial design for this indication Infliximab is best supported by evidence, but was superior to placebo for the endoscopic, not clinical endpoint Drugs with tenuous evidence base (metronidazole, thiopurines, aminosalicylates) remain widely used No controlled studies of biologics other than TNF antagonists Little data to guide treatment decisions after TNF antagonist failure	Development of regulatory guidance incorporating and endoscopic primary endpoint Performance of controlled trials for biologics other than TNF antagonists with particular emphasis on patients after TNF antagonist failure

## Data Availability

Not applicable.

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
