# Peer review of "Contemporary Management of Postoperative Crohn’s Disease after Ileocolonic Resection"

_jcm, 2022, doi:10.3390/jcm11226746_

Round 1

Reviewer 1 Report

This timely and well written review from Hanzel et al. provides a useful overview of current knowledge about management of postoperative Crohn’s disease.

However, the terms “((post-operative[Title/Abstract]) OR (postoperative[Title/Abstract]) OR (resection[Title/Abstract])) AND ((Crohn[Title/Abstract]) OR (Crohn's[Title/Abstract]))” used in PubMed search may exclude important and relative articles with similar terms such as “postsurgical” or “recurrence”. Authors could consider adding more similar terms in their search and revise their manuscript accordingly. For example, an important metanalysis on diagnostic accuracy of intestinal ultrasound in Crohn’s disease recurrence named “Diagnostic Accuracy of Ultrasonography in the Detection of Postsurgical Recurrence in Crohn's Disease: A Systematic Review with Meta-analysis “ by Rispo A et al is excluded by the search.

 Apart from that, I could also suggest minor revisions:

1.     A lot of references in the section “2. Diagnosing recurrence in postoperative Crohn’s disease” are not numbered correctly which complicated the review process. I suggest the authors would correct the reference numbers accordingly.

2.     In Table 1 some rows are not shown correctly. Please consider revising.

3.     In line 260, I suggest changing “provided that the percent-259 age of ulcerated or affected surface was below 10% and 30%, respectively” to “… below 10% and 50%, respectively”, in order to be accordant to SES-CD parameters.

4.     In line 336, please consider revising “This is the first MRE designed specifically for postoperative CD”. (I suggest: “This is the first MRE index designed specifically for postoperative CD”)

5.     In Table 3, authors may consider revise the columns width and text length. Shorter text and wider columns may be easier for readers to read.

Author Response

We thank the reviewer for their thoughtful comments.

  1. Authors could consider adding more similar terms in their search and revise their manuscript accordingly.

We have expanded the search strategy to avoid omissions and ensure that the review is comprehensive. The updated search yielded approximately 1200 additional search results which were examined. The Rispo meta-analysis was picked up by the original search because of its abstract "The postoperative course of Crohn's disease (CD) is best predicted by ileocolonoscopy."

2.     A lot of references in the section “2. Diagnosing recurrence in postoperative Crohn’s disease” are not numbered correctly which complicated the review process. I suggest the authors would correct the reference numbers accordingly.

All citations in Section 2 were checked manually and revised where applicable. To improve legibility and avoid conflicts with the reference manager, this was done without tracking changes.

3.     In Table 1 some rows are not shown correctly. Please consider revising.

Table 1 has been revised to ensure coherence between lines and columns.

4.     In line 260, I suggest changing “provided that the percent-259 age of ulcerated or affected surface was below 10% and 30%, respectively” to “… below 10% and 50%, respectively”, in order to be accordant to SES-CD parameters.

Thank you for picking up our error, which has now been corrected.

5.     In line 336, please consider revising “This is the first MRE designed specifically for postoperative CD”. (I suggest: “This is the first MRE index designed specifically for postoperative CD”)

Corrected

6.     In Table 3, authors may consider revise the columns width and text length. Shorter text and wider columns may be easier for readers to read.

Table 3 has been widened to facilitate reading

Reviewer 2 Report

thank you for let me reviewing this paper. I appreciate the extensive review of the literature and current clinical practice. My only concern is about the title, since the author do not actually propose any advances in the management of CD, but they provide an overview of diagnosis and treatment after ileocolonic resection. Furthermore, I would definitely change the term Postoperative Crohn's Disease that is too vague. In fact, CD patients might be operated on by ileocecal resection (that is the topic of the Author's review), but also by colonic resections, multiple jejune-ileal stricturepasty, duodenal involvement, or perianal disease.  

I feel it could be better to change the title, for example: management of postoperative cd after ileocecal resection. 

Author Response

We have amended the title to "Contemporary Management of Postoperative Crohn’s Disease after Ileocolonic Resection" - this reflects the subset of patients after surgery for CD addressed by the review and the fact that we document contemporary practices (shift towards noninvasive monitoring etc) which may not necessarily by viewed as advances.

Round 2

Reviewer 1 Report

I would like to thank authors for their prompt attention to reviewer's comments and their reply. I have no more comments.

Author Response

Thank you.